# Towards a robust experimental framework and benchmark for lifelong language learning

**Aman Hussain**
ILLC, University of Amsterdam
email@amanhussain.com

**Nithin Holla**
Amberscript
nithin.holla7@gmail.com

**Pushkar Mishra**
Facebook AI
pushkarmishra@fb.com

**Helen Yannakoudakis**
Dept. of Informatics, King's College London
helen.yannakoudakis@kcl.ac.uk

**Ekaterina Shutova**
ILLC, University of Amsterdam
e.shutova@uva.nl

## Abstract

In lifelong learning, a model learns different tasks sequentially throughout its lifetime. State-of-the-art deep learning models, however, struggle to generalize in this setting and suffer from catastrophic forgetting of old tasks when learning new ones. While a number of approaches have been developed in an attempt to ameliorate this problem, there are no established, unified or generalized frameworks for rigorous evaluations of proposed solutions; a problem which is particularly pronounced in the domain of NLP. The few existing benchmarks are typically limited to a specific flavor of lifelong learning – continual open-set classification – where new classes, as opposed to tasks, are learned incrementally. Moreover, the only general lifelong learning benchmark combines a multi-label classification setup with a multi-class classification setup resulting in misleading gradients during training. We empirically demonstrate that the catastrophic forgetting observed here can be attributed to the experimental design rather than to any inherent modeling limitations. To address these issues, we propose an experimental framework for true, general lifelong learning in NLP. Using this framework, we develop a comprehensive suite of benchmarks that target different properties of lifelong learning (e.g., forgetting or intransigence); experiment with diverse facets of language learning: multi-domain, multilingual, and different levels of linguistic hierarchy; and present a continuous evaluation scheme under a new metric: *Area Under the Lifelong Test Curve*. Our framework reveals shortcomings of prevalent memory-based solutions, demonstrating they are unable to outperform a simple experience replay baseline under the realistic lifelong learning setup.

## 1 Introduction

Research has shown that modern deep learning methods are able to learn a variety of complex language tasks such as text classification and document ranking [57], language modeling and common sense reasoning [6], question answering and document summarization [60]. In a single-task learning setting, these models can learn a given task well and achieve state-of-the-art performance. However in the lifelong learning setting, where several tasks are learned in sequence, such models often fail to generalize and suffer from *catastrophic forgetting* [35], forgetting old tasks when learning new ones. While there is a substantial ongoing effort to solve this problem, existing work has shown that simple baselines can outperform a multitude of recently proposed solutions [39], pointing that, despite existing progress, current methods are far from solving lifelong learning. Crucially, for substantial progress to be made, there is a need for established consensus and/or unified and

35th Conference on Neural Information Processing Systems (NeurIPS 2021) Track on Datasets and Benchmarks.

standardized approaches to evaluating lifelong learning. Parisi et al. [37] note that considerably less attention has been paid to the rigorous evaluation of such solutions, while Farquhar and Gal [17] have furthermore demonstrated a number of issues with existing lifelong learning experimental settings, showing that existing evaluations are biased towards prior-focused (regularization) approaches. They discover several blindspots in the existing evaluations that have created misleading comparisons in recent works. Specifically, approaches to lifelong learning often adopt assumptions that simplify the problem by assuming access to explicit task identifiers [27, 46] or by allowing unrestricted access to the entire dataset of the current task to perform multiple passes over the data [42]. However, such assumptions do not extend to general lifelong learning which must be able to adapt and generalize to new unseen tasks in a realistic setting that precludes the use of any such explicit cues.

A recent survey [4] highlights that the dearth of dedicated datasets and benchmarks for evaluating lifelong learning is particularly pronounced in the domain of natural language processing (NLP). The few existing benchmarks in NLP are typically limited to a specific flavor of lifelong learning – continual open-set classification [47]. They are predominantly focused on sequentially learning new classes from a single task or very similar tasks. However, general lifelong learning must incorporate truly diverse tasks if it is to be robust, versatile, and represent a holistic approach to learning. Particularly within the NLP domain, the majority of existing benchmarks are not general in scope. d'Autume et al. [15] present an exception by building a benchmark for the general lifelong learning scenario: learning multiple tasks without explicit task identifiers. However, a shortcoming we identify is the lack of distinction between different learning settings (multi-label vs. multi-class classification), resulting in misleading gradients during training, which we show directly leads to catastrophic forgetting (Section 3). Furthermore, we find the benchmark accidentally leaks explicit task identifiers, and experimentally demonstrate that, once task identifiers are given, a simple system that utilizes multiple task-specific heads outperforms the top-performing system on this benchmark.

Therefore, we consider lifelong learning in NLP in its true, generalized form, i.e., distinct multiple tasks without giving away task identifiers explicitly. Specifically, we choose to work with pre-trained, contextualized language models considering they have been a mainstay of this field. Our contributions are as follows: **1)** We analyze existing and prevalent lifelong learning benchmarks in NLP, and present and discuss shortcomings in their design towards truly generalized lifelong learning; **2)** We propose *Degree-of-Belief*, a novel experimental framework that facilitates a general lifelong learning setting for language models by incorporating multiple tasks without explicit task identifiers – Figure 3 illustrates *Degree-of-Belief* whereby the model must state its belief in the truth of a statement given the context and its past knowledge; **3)** We extend our framework with a comprehensive suite of benchmarks that target different properties of lifelong learning (e.g., forgetting, intransigence) as well as multiple facets of language learning: multi-domain, multilingual and different levels of linguistic hierarchy; **4)** We propose a new metric, *Area Under the Lifelong Test Curve*, that allows for continuous model evaluation by measuring test accuracy throughout the lifelong learning process; **5)** We release an open-source Lifelong Learning Library[1] to evaluate proposed solutions to general lifelong learning and investigate different properties of lifelong learning using a suite of data streams (our framework can be easily extended with new data streams and tasks); **6)** We present and evaluate a number of baselines on our benchmarks and show that prevalent memory-based solutions are unable to outperform simple experience replay models in a general lifelong learning setup.

## 2    General Lifelong Learning

In supervised learning, our aim is to learn a model $f : \mathcal{X} \rightarrow \mathcal{Y}$ to predict the target $\mathcal{Y}$ given the input $\mathcal{X}$ using a dataset $D_{tr} = \{(\mathbf{x_i}, \mathbf{y_i})\}_{i=1}^{n}$ of $n$ examples, each consisting of an input vector $\mathbf{x_i} \in \mathcal{X}$ and a target vector $\mathbf{y_i} \in \mathcal{Y}$. The examples are *iid* (independently and identically distributed), assumed to be independently sampled from a fixed distribution $P_T$ which characterizes the target task. Conversely in lifelong learning, we observe a data stream of $n$ examples $\{(x_1^1, y_1^1), \ldots, (x_i^j, y_i^j), \ldots, (x_n^m, y_n^m)\}$ sampled from a sequence of $m$ different task distributions $\{P_1, \ldots, P_j, \ldots, P_m\}$. The examples are not sampled from identical distributions and the sampling process is dependent on the sequence of tasks. Therefore, the *iid* assumption is violated during lifelong learning, posing a much more challenging setting than "standard" single-task learning. During lifelong learning, models often suffer from catastrophic forgetting where performance on previously seen tasks drops precipitously as new tasks are learned.

---

[1] `https://amanhussain.com/lifelong-learning/`

Herewith, we lay the following desiderata and define lifelong learning within a set of predefined conditions that must simultaneously hold for it to be considered truly 'general':

**A: Task plurality** A lifelong learning model must learn multiple different *tasks* sequentially. This is not to be conflated with open-set classification where new unseen *classes* are learned incrementally instead. Continual open-set classification and open-world recognition have a flavor of lifelong learning [18], but they are essentially classification tasks.

**B: Task generality** A lifelong learning model must generalize to new unseen tasks without requiring explicit task identifiers. The use of explicit task identifiers oversimplifies the problem as one can directly train a separate model for each task. When models are trained separately, there is no helpful transfer of knowledge since the weights are not shared at all. Correspondingly, there is also no catastrophic interference when there are no shared model parameters. The transfer-interference trade-off is discussed in detail by Riemer et al. [43].

**C: Online stream** A lifelong learning model does not have access to the training data of previously seen tasks; otherwise, it can simply re-learn from past data streams to ameliorate catastrophic forgetting. It may, however, have capacity to save past data in some (compressed) form (e.g., a memory component), while simultaneously being constrained by the fourth desideratum below. In practice, there may be substantial computational cost involved in data retrieval, which can limit its use. Also, a model that re-learns previously seen tasks in short intervals using a high retrieval rate may start to resemble multi-task learning, diverging from the lifelong learning paradigm (Section 4.4).

**D: Space complexity** A lifelong learning model must have bounded space complexity where, for example, the growth of model parameters and available memory is capped. An unconstrained model memory may lead to learning settings where all previously seen data is directly stored and maintained by the model in some (compressed) form. Lifelong learning is not required if all the training data can be stored and re-trained upon.

The desiderata proposed above are in close alignment with several lifelong learning papers [17, 4, 30, 39], thereby establishing consensus on what general lifelong learning should look like.

## 3 Related Work and Limitations

In general, solutions proposed for lifelong learning can be classified into the following categories: i) replay based approaches [44, 42, 50, 33, 9]; ii) regularization based approaches [27]; iii) architecture based approaches [46, 59, 54]. In the domain of language, a number of lifelong learning methods have also been proposed, including embedding aligned episodic memory replay [53]; memory-based parameter adaptation with sparse experience replay (MbPA++) [15]; meta-learning with sparse experience replay [23]; and language modeling for lifelong language learning [52].

However, we lack a unified and standardized approach for evaluating and benchmarking these solutions. While some of the recent works, e.g. Sun et al. [52], have adopted multi-task benchmarks such as decaNLP [34], others [58] have designed their own experiments to study the specifics of the problem. Kruszewski et al. [28] propose CALM, a lifelong learning benchmark of character- and word-based language modeling tasks. d'Autume et al. [15] present the Lifelong Text Classification benchmark which consists of five different text classification tasks adopted from Zhang et al. [64]: AG News article classifcation (4 classes), DBPedia ontology classification (14 classes), Yahoo Answers topics classification (10 classes), sentiment analysis on the Yelp and Amazon reviews data (5 classes each). They additionally present the Lifelong Extractive Question Answering benchmark comprising three datasets: SQuAD 1.1 [41], TriviaQA [24], and QuAC [10]. Wang et al. [53] present two different benchmarks: Lifelong FewRel, a lifelong few-shot relation classification dataset [21] separated into 10 disjoint clusters of relation classes; and Lifelong Question Relations, a lifelong single-relation question–answer dataset [5] with 20 disjoint clusters. Models learn to classify the relations from each disjoint cluster, as they are presented sequentially.

**Limitations of existing work** include: **1) Lack of task plurality:** Barring the Lifelong Text Classification benchmark, none of the datasets include multiple different tasks. Lifelong Extractive QA [15] concerns extractive question–answering task on different datasets, primarily targeting multi-domain lifelong learning; while CALM [28] only concerns language modeling tasks. The Lifelong FewRel and Lifelong Question Relations datasets [53] focus on classifying relations within each disjoint cluster as a task on its own. This resembles continual open-set classification rather than general

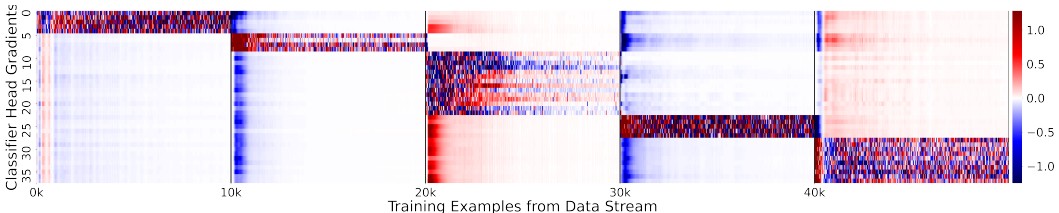

Figure 1: Gradients for each class on the shared classifier head when training on the stream of {Yelp Reviews, AG News, DBPedia, Amazon Reviews, Yahoo Answers Topics} with 10k examples each.

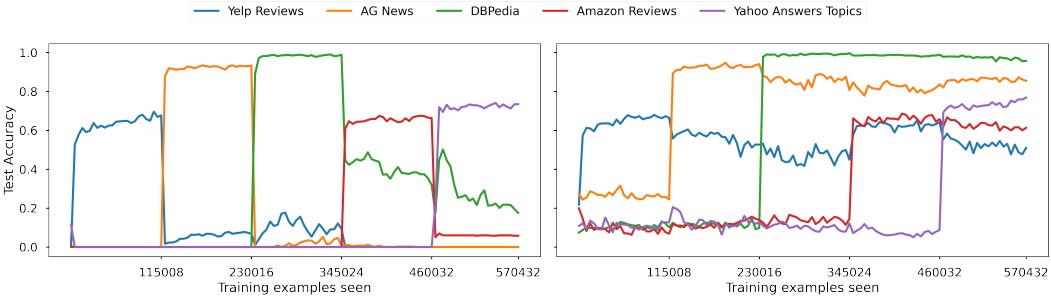

Figure 2: Left: test accuracy of each task while training a shared classifier head; right: test accuracy of each task while training multiple classifier heads (one for each task).

lifelong learning in that it involves the single task of relation detection, albeit over an increasing number of relation classes. All the above datasets are limited in scope in that they adopt a narrow definition of a "task". **2) Lack of task generality:** The Lifelong Text Classification benchmark [15] meets most of the desiderata for general lifelong learning. It defines no explicit task identifiers and incorporates task plurality to an extent, as it comprises two different tasks: sentiment analysis and topic classification (although both fall under text classification). The dataset consists of 38 classes in total; although the authors reduce the number to 33 by treating the class labels from Amazon and Yelp reviews the same. We conduct an analysis of this benchmark and experimental setup, keeping the 38 classes separate for a more generalized setting. Following the authors' original work, we train a shared classifier head on top of pre-trained BERT [14] such that it learns to classify the classes from each task in sequence. In effect, the model generates a probability distribution over $N$ classes given an input text $x$ and model parameters $W$: $p_i = P(y = i|x, W) \quad \forall i \in \{1, 2, \ldots, N\}$ where $N = 38$. It is trained using cross-entropy loss: $-\sum_{i=1}^{N} t_i \log(p_i)$, where the true probability distribution is represented by a one-hot truth vector $\mathbf{t} \in \mathbb{R}^N$. It is thus framed as a multi-class classification problem.

However, we identify two issues with this experimental setup. First, an input text can belong to more than one class and, therefore, a more appropriate learning setting would be multi-label as opposed to multi-class classification; for example, a news article can belong to the "politics" topic *and* have a "negative" sentiment. However, the one-hot truth vector zeros out all other labels except the target one, therefore presenting an incomplete truth distribution. When training on new tasks sequentially, the model is penalized for giving any amount of probability mass to classes outside the current task.

We verify this empirically by training the model described above on 10k examples from each dataset using order (i) from [15]: Yelp reviews, AG News, DBPedia, Amazon reviews and Yahoo answers topics. During training, we record the gradients received by the classifier head weights. In Figure 1, we see the aggregated gradients (y-axis) on the shared classifier head across the training iterations (x-axis). The gradients have been summed across the hidden dimension of the head such that the aggregate reflects the total gradient for each of the 38 classes at every training iteration. The vertical lines after every 10k examples mark the task boundaries. When the model is being trained on a particular task, the class heads of that task receive diverse and informative gradients. When training on a new task begins at the task boundaries, the class heads of the previous task start receiving

uniform and uninformative gradients. These gradients might reverse all the learning on the previous task since they are based on incomplete ground truth labels. The largest deterioration in performance is observed immediately after the tasks are switched, i.e., right after the task boundaries. On the left of Figure 2, we plot the train-time test accuracy of each task when using a shared classifier head. When a new task is introduced, the accuracy on that task increases, whereas the accuracy on previous tasks drops precipitously, demonstrating the patterns of catastrophic forgetting during training. However, the drops in the accuracy of previous tasks are rather associated with the strength of the misleading / incomplete gradients on the class heads of other tasks.

The second issue we identify with the experiment setup is that it inadvertently leaks explicit task identifiers. All the classes, except those from the current task, are always set to zero. Thus, during training, we can directly exploit this information to infer the current task identity. For example, we can use the labels trick [62] to train only the heads which correspond to labels that exist in the current batch; thereby training each of the task-specific heads separately. Ultimately, if the task identity is known, we can use multiple heads, one for each task, instead of a shared head. In Figure 2, we compare the shared head setup (left) against the multiple head one (right). For a direct comparison with the results in d'Autume et al. [15], we train on the datasets (115k training examples each) using order (i) and five different seeds: [1,2,3,4,42]. As hypothesized, we see no major catastrophic forgetting in the multiple head setting (the average test accuracy is $72.68\% \pm 0.005$). Therefore, multiple heads alone can get very close to the multi-task learning upper bound (73.7%) achieved in d'Autume et al. [15], when task identifiers are present or can be inferred. This suggests that almost all of catastrophic forgetting in this benchmark seems to stem from the adopted experimental setup.

One may argue that switching to a multi-label classification setting would present an immediate solution. However, we are practically constrained by the availability of data annotated across sets of different tasks. Therefore, in the shared setup, it is impractical to incorporate multiple tasks without additional annotations. Another way to incorporate multiple tasks would be to have multiple task-specific heads, so as to avoid misleading gradients on other task heads. However, in this case, the model will need explicit task identifiers to know which head to use, leading to loss of task generality. To ensure a general lifelong learning setup, we need to implement the desiderata of both task plurality and task generality without explicitly or indirectly leaking task identifiers.

## 4    Experimental Framework

### 4.1    Implementation of desiderata

We propose *Degree-of-Belief*, a framework that satisfies the task plurality and task generality desiderata for general lifelong learning. In this framework, the model should state its belief in the truth of a statement given a context and its past knowledge. Formally, we model the probability of a statement $x$ being true as: $P(y = 1|x, c, W)$, where the context $c$ is some input text, the statement $x$ is some assertion about that text, and the past knowledge is expressed by the learned weights $W$ of the model.

Multiple different tasks can be encoded into this framework by using textual task descriptions in the statement $x$ itself. Instead of using explicit task identifiers which need to be hard-coded manually, the model is required to implicitly understand task descriptions. Consider, for example, the task of natural language inference, where the context may be formed by concatenating the hypothesis sentence after the premise sentence, and example statements can take the form of: "This implies entailment" or "It is a contradiction". To ensure that the implicit task identifiers cannot be hard-coded or memorized, we use semantically similar but syntactically different statements when encoding a task in our framework; e.g., for word-in-context classification, the context is created by joining two sentences that use the same word $w$ but in a different sense, and the statement is formed by randomly choosing either one of these two templates: "$w$ is used in the same sense" or "$w$ is the polysemous word". The training data stream, however, comprises of both true and false statements about a context, such that the model has both positive and negative examples to learn from. The false statements are formed by using incorrect labels for classification tasks or partially corrupting the token labels for structured prediction tasks. The Datasheet B in the appendix provides the detailed processing steps to encode all the different tasks. Figure 3 presents an example instantiation of our degree-of-belief framework, exemplifying the statement, context and truth label for five different tasks.

**Limitations**    The degree-of-belief framework is designed such that it can cover a diverse set of language understanding tasks (such as inference, parsing, multi-choice QA, and relation extraction);

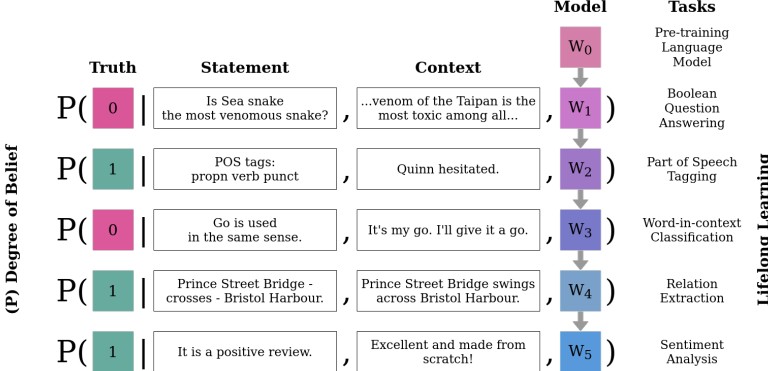

Figure 3: Example instantiation of degree-of-belief for general lifelong learning with pre-trained language models, demonstrating the statement, context, and truth labels for our stream of 5 tasks.

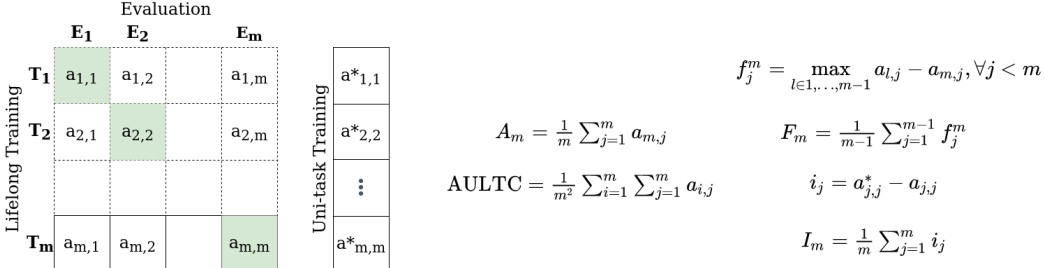

Figure 4: Lifelong learning evaluation metrics: forgetting $F_m$; intransigence $I_m$; final average accuracy $A_m$; area under the lifelong test curve (AULTC). The accuracy matrix (left) records the evaluation accuracy $a_{i,j}$ on task $j$ when training on task $i$. The single task accuracy vector records the evaluation accuracy $a_{i,i}^*$ on task $i$ when training a single model for each task $i \in 1, \ldots, m$.

however, a limitation is that, in its current form, it does not cover generation and retrieval tasks. One framework that can cover the full range of diverse and distinct NLP tasks presents a challenge to the research community: among others, there is a need for architectures that fulfill the desiderata of task generality by eschewing task-specific implementations, while tasks such as generation pose an additional challenge in the requirement of objective evaluations of text quality and/or reference texts [7], conflating the challenge of learning and evaluating the task with the challenge of catastrophic forgetting in lifelong learning. We hope that degree-of-belief will serve as a platform for future work in lifelong learning that seeks to extend and adapt it to cover additional types of NLP tasks, drawing inspiration from work such as Text2Text [40] and the Multitask QA framework [34].

## 4.2 Evaluation scheme

We follow the work of Chaudhry et al. [8] and evaluate lifelong learning models on this set of metrics: average accuracy $A_m$, forgetting $F_m$, and intransigence $I_m$ (see Figure 4). Forgetting $F_m$ is defined as the difference between the maximum performance achieved on a task during lifelong learning and the final performance on the task. Intransigence $I_m$ is defined as the inability to learn new tasks during lifelong learning. The average accuracy defines the final performance of the model – it is measured at the end of lifelong learning when the model has been trained on all tasks from 1 to $m$. As lifelong learning is a continuous process, we extend this set and propose an online metric that measures performance continuously throughout learning – area under the lifelong test curve (AULTC). Rather than solely measuring average accuracy at the end of the training process, this metric allows us to assess the average accuracy of the model across all $m$ tasks after every $b$ training iterations, where the frequency of evaluation $b$ controls the metric's level of granularity; $b$ can be varied depending on computational constraints. On plotting the average accuracy of the model throughout training, we get the lifelong test accuracy curve representing the overall lifelong learning performance.

## 4.3 Benchmark design

To design our benchmark, we consider a total of 16 different datasets encompassing 10 distinct tasks (Appendix B). Among these, we seek to identify tasks for which pre-trained language models like BERT [14] have been shown to achieve good (single-task) performance after task-specific fine-tuning. We check for this criteria to avoid conflating the challenge of learning the task with the challenge of catastrophic forgetting in lifelong learning. Specifically, we find the following 10 datasets to meet this criterion: part-of-speech tagging (UDPOS) [61], named entity recognition (PANNER) [36], few-shot relation extraction (FewRel) [22], Word-in-Context (WiC) [38], boolean question answering (BoolQ) [11], DBPedia ontology classification [29], Amazon Reviews sentiment analysis [25], AG News article classification, Yahoo Answers topics classification and Yelp Reviews sentiment analysis [65].

**Overall performance:** We start by considering two different settings based on the number of tasks in the data stream. For our default **Standard Stream**, we randomly select 5 datasets: UDPOS, FewRel, WiC, Yelp Reviews and BoolQ; for our **Long Stream**, we use all of the 10 datasets and arrange them in a random order. In order to identify the most challenging sequence of tasks for the **Standard Stream**, we measure the performance of all its possible permutations (120) on all the evaluation metrics in Section 4.2. Figure 6 in Appendix A.1 presents the distribution of each evaluation metric across all permutations. Based on this, we select the order for the **Standard Stream** as: BoolQ, UDPOS, WiC, FewRel and Yelp Reviews. The task order permutation above has the minimum AULTC (0.5872), the minimum final accuracy $A_m$ (0.6294), and the highest forgetting $F_m$ (0.1677). While this stream does not have the highest possible intransigence ($I_m = 0.0338$ vs 0.0607), we nevertheless select it since it presents the most challenging order overall. To keep the experiment runtime manageable (see Appendix A.4), we use 10k randomly sampled training examples per dataset in all the data streams (we find that 10k is sufficient for our tasks to achieve good single-task performance; Table 9). However, we further seek to study training set size effects and additionally design a **Large Stream** and a **Larger Stream**. These are the same as **Standard Stream** but with 50k and 100k training examples per dataset respectively.

**Metric-specific performance:** To further specialize our study, we identify task orders at the level of each individual metric. Specifically, we identify the order that displays the highest intransigence since the **Standard Stream** does not show this behavior. However, for the other four metrics, we want to capture the most frequent behavior since the **Standard Stream** already records their worst possible numbers. Therefore, we find the most frequent value of each metric and select any one of the orders that falls in that frequency bin. The resulting streams targeting each metric specifically are: **Forgetting Stream**: FewRel, UDPOS, WiC, Yelp Reviews and BoolQ; **Intransigence Stream**: Yelp Reviews, UDPOS, WiC, BoolQ and FewRel; **Final Accuracy Stream**: WiC, UDPOS, BoolQ, FewRel and Yelp Reviews; **AULTC Stream**: UDPOS, WiC, FewRel, Yelp Reviews and BoolQ.

We extend our suite of benchmark streams even further such that it can assess disparate and complementary facets of language understanding: multi-domain learning, multilingual learning, and learning at different levels of linguistic hierarchy.

**Multi-domain and multilingual:** To study multi-domain lifelong learning, we design two different streams. In the first, **MultidomainA**, we take the notion of 'domain' to the extreme and focus on five distinct text classification tasks: Yelp Reviews, AG News, DBPedia, Amazon Reviews and Yahoo Answers Topics; in the second, **MultidomainB**, we focus on task-specific domain variance and adopt the Multigenre Natural Language Inference dataset [55]. We select and order the most frequent NLI genres / domains in the data stream as follows: Fiction, Government, Slate, Telephone and Travel. For multilingual lifelong learning, we propose two different streams where we keep the task fixed and vary the language. We select languages from five different language (sub)families (Semitic, Indo-European, Turkic, Uralic, Sino-Tibetan) to construct two different streams: **MultilingualA**, tackling part-of-speech tagging (UDPOS) across the following (random) order of languages: Arabic, Hindi, Turkish, Finnish, (traditional) Chinese; **MultilingualB**, focusing on named entity recognition (PANNER) across the following (random) order: Arabic, Bengali, Turkish, Finnish, Chinese.

**Linguistic hierarchy:** To study lifelong learning across different levels of linguistic hierarchy, we select the following five tasks in order of increasing complexity: part-of-speech tagging (UDPOS), named entity recognition (PANNER), Word-in-Context classification (WiC), co-reference resolution (WSC) [31] and relation extraction (FewRel).

### 4.4 Baselines and comparison systems

We implement a set of strong baselines for our benchmark suite. We use the BERT-base architecture (mBERT for multilingual streams) with a binary classification head on top. The Transformers library [56] provides the pre-trained weights and the default hyperparameters. The learning rate is set to $3 \times 10^{-5}$ and the batch size to 25. Following our online stream desideratum, all training is done in a single pass of the data stream. To control for computational complexity, we evaluate after every 500 training examples ($b = 500$) and limit the test set size of each task to 1000. Our test set is randomly sampled from the full test set and maintains the original class distribution. However, in Appendix A.7, we present additional experiments where we empirically validate that our conclusions on our test set (for the **Standard Stream**[2]) match those on the full set of test examples.

**Multi-task and single-task learning:** In multi-task learning, the model learns multiple tasks simultaneously as opposed to sequentially. The multi-task training examples are randomly sampled from the task distributions $P_T$ in no particular order. Given that the *iid* assumption holds in this setting, it serves as a strong point of reference against *non-iid* lifelong learning. Further inspired by the effectiveness of explicit task identifiers, we train single-task learning models on each task individually. As strong equivalents of achievable task-specific performance, we compare their final average accuracy against the lifelong learning counterpart.

**Experience replay:** Experience replay is another way to approximate the *iid* property in the data stream. In experience replay, the model maintains a memory buffer, and stores the examples seen during training, each with a probability $P_{write}$. At specific intervals during training, the model retrieves some of the stored examples and re-trains on them. The replay interval $R_{interval}$ [23] is defined as the number of examples the model has seen between two successive draws from memory. The replay rate is defined as $R_{rate} = N_{replay}/R_{interval}$, where $N_{replay}$ is the number of examples sampled from memory. When $P_{write}$, $R_{interval}$ and $R_{rate}$ all equal to 1, the learning setting transforms into a multi-task one. For experience replay, we use the following hyperparameters: $P_{write} = 0.1, R_{interval} = 1000, N_{replay} = 100$, thereby having a replay rate $R_{rate}$ of 10%.

## 5 Experiments and Discussion

**Lifelong learning baselines:** We evaluate all baseline methods discussed in Section 4.4 on all the streams described in Section 4.3. Tables 2 and 3 in the appendix present the full set of results for all metrics in our evaluation scheme (Section 4.2). Comparing the **Standard Stream** against the **Large Stream** and **Larger Stream**, we observe that, for lifelong learning, when the number of training examples per dataset is increased, Forgetting and Intransigence decrease, while AULTC and Final Accuracy increase. A similar pattern is observed when increasing the number of tasks in the **Long Stream** where, again, Forgetting and Intransigence decrease, and AULTC and Final Accuracy increase. However, when lifelong learning performance is compared against the multi-task/single-task learning models within each stream, we find that the relative performance difference remains largely the same as in the Standard Stream (Table 5). On the one hand, this indicates that more data alone is not enough to ameliorate catastrophic forgetting; on the other hand, the results present evidence that longer streams of data may not necessarily deteriorate lifelong learning performance if related but diverse tasks exist within the same (long) stream (e.g., includes 3 topic classification tasks and 2 sentiment analysis ones). In the **Linguistic Stream** (Table 2), we find that lifelong learning performance is again substantially lower compared to multi-task and single-task learning. This indicates that curriculum learning, where tasks are ordered by increasing complexity [2], may also suffer from catastrophic forgetting. In contrast to the above, however, we observe negligible Forgetting in both the **MultilingualA** and **MultidomainB** streams. For **MultidomainB**, this can be explained by the fact that it consists of one task (NLI) across different domains, whereas **MultidomainA** consists of different tasks (topic classification, sentiment analysis) from different domains. In contrast, **MultilingualA** tackles part-of-speech tagging, whereas **MultilingualB** focuses on named entity recognition which can be particularly language specific. The results suggest that catastrophic forgetting can be more or less pronounced depending on the given data stream and its characteristics. Overall, we observe that the AULTC of multi-task learning is much higher than that

---

[2]We primarily use this stream as runtime increases prohibitively with the increase in test set size – 2 hours to run an experiment using our test set vs 51 hours using the full version.

of lifelong learning, with **MultilingualA** being the only exception. For Final Accuracy, we observe the following performance: single-task $\geq$ multi-task $\geq$ lifelong learning, except for **MultidomainB**.

Overall, we observe that lifelong learning often shows lower intransigence than multi-task learning (e.g., **Linguistic** and **AULTC** stream; Tables 2 and 3). This suggests that the lifelong learning models are more adaptable to learning new tasks than multi-task ones. As expected, we also observe that experience replay reduces Forgetting whenever it is used by the model. Even though replay serves as a simple baseline, it is particularly effective. However, our online stream desideratum limits the replay interval and number of replay examples in experience replay. The space complexity desideratum also discourages the use of experience replay due to the linear growth in its space requirements.

Finally, we perform two further analyses: 1) we examine performance on the minority classes and less representative examples, and plot the F1 scores and true positive/negative rates on the 'true' class (where the input statements are true; Figure 3) throughout lifelong learning on the **Standard** stream (Appendix A.6). Overall, we observe the true positive rate suffers given the majority of examples belong to the 'false' class. The F1 scores for the 'true' class/examples drop, sometimes rapidly, as new tasks arrive. 2) We test whether explicit cues in implicit task identifiers are memorized by the model (even though we devise semantically similar but also syntactically different statements), and run an experiment where we increase the number of different statements when encoding a task (Appendix A.8). In the Standard stream, we remove the punctuation marks from some of the statement templates of the BoolQ, UDPOS and FewRel tasks, and replace keywords such as 'positive' with 'good' and 'negative' with 'bad' in some of the Yelp Review task statements. We find this does not affect our conclusions and observed patterns, suggesting that the model does not rely solely on specific cues in the input implicit statements when attempting to identify tasks and make predictions.

**Task-specific learning and identification:** As discussed in Section 4.1, it is desirable for the model to learn to understand implicit task descriptions and make predictions accordingly. One way in which we can investigate whether the model has learned to identify tasks is by inspecting its parameter space and, specifically, whether it resorts to using a subset of its parameters to learn and make predictions on the identified task. To quantitatively assess this, we measure the gradient overlap throughout lifelong learning. At every train step $t$, we store the set of model parameters that receive non-zero gradients as $S_t$. These set of parameters form a sub-network that is actively learning at that step $t$. When a new task arrives, we expect the model to recognize the change and switch to a different sub-network. The intersection or overlap between the old and the new sub-network should depend on the similarity between the old and the new task. To measure this overlap, we take the set intersection of $S_t$ with $S_{t-1}$. In Figure 5, we plot the gradient overlap during lifelong learning and multi-task learning on the **Standard Stream**. For lifelong learning, we observe that gradient overlap stays roughly the same when learning a single task but it changes drastically when the task changes. We observe small fluctuations in the gradient overlap for multi-task learning since the model is learning multiple different tasks simultaneously (alternate training). Overall, we find that the lifelong learning model is able to recognize the change in tasks and utilizes diverse parameter subsets for these. In Appendix A.9, we plot the gradient overlap for all remaining data streams in our benchmark. We consistently observe the same behaviour but at varying degrees, depending on the task similarities in the data stream. Although gradient overlap may be viewed as a coarse measure of assessment, our results suggest that the model is able to identify tasks and use its parameter space accordingly.

**Memory-based vs experience replay:** We now turn to experiments where we use our benchmark to study prevalent memory-based solutions to lifelong learning. Specifically, we experiment with Average Gradient Episodic Memory (A-GEM) [9], and Memory-based Parameter Adaptation with replay (MbPA++) [15]. A-GEM aims to prevent the loss on previous tasks from increasing by solving a constrained optimization problem based on the gradients from examples of past tasks in the memory. While A-GEM, by design, needs explicit task identifiers to retrieve samples from memory, we adapt it to work within our realistic framework by randomly sampling from its memory instead. MbPA++ performs local adaptation on K-nearest neighbors from the memory during inference while carrying out experience replay during training. For MbPA++, we use pre-trained BERT base as our key network. We set the number of neighbours $K = 25$ and number of local adaptation steps to $L = 10$.

These two methods are specifically selected since they are designed as an advancement over experience replay. We, therefore, compare them against our experience replay baseline ($R_{rate} = 10\%$) to see how they improve upon it. For both A-GEM and MbPA++, we use the same experience replay hyperparameters for a fair comparison. We run the experiments with five random seeds: $[1, 2, 3, 4, 42]$.

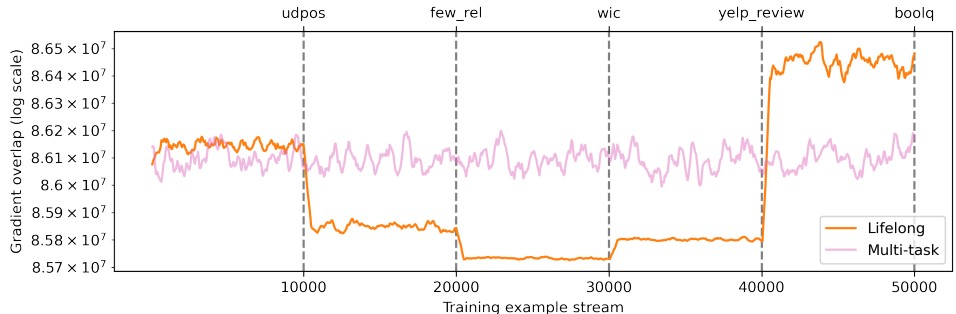

Figure 5: Gradient overlap during lifelong learning and multi-task learning on the Standard stream. The y-axis shows the number of parameters that are shared between two consecutive sub-networks, i.e., parameters which received non-zero gradients at time $t$ and $t-1$.

Table 1: Comparison of memory-based solutions on the **Standard stream**.

|  | **AULTC** | **Forgetting** | **Intransigence** | **Final Accuracy** |
|---|---|---|---|---|
| Lifelong | $60.95 \pm 1.64$ | $12.78 \pm 3.30$ | $1.52 \pm 1.70$ | $68.26 \pm 3.87$ |
| Replay 10% | $62.81 \pm 2.14$ | $4.35 \pm 2.13$ | $1.82 \pm 2.95$ | $76.21 \pm 5.08$ |
| A-GEM | $61.27 \pm 1.48$ | $6.93 \pm 2.48$ | $1.56 \pm 1.84$ | $74.30 \pm 2.59$ |
| MbPA++ | $62.93 \pm 3.20$ | $4.21 \pm 2.98$ | $1.79 \pm 3.67$ | $76.45 \pm 5.11$ |
| Single task | – | – | – | $80.64 \pm 1.37$ |
| Multi-task | $74.34 \pm 0.30$ | $3.95 \pm 0.74$ | $6.17 \pm 1.37$ | $77.72 \pm 0.89$ |

Table 1 presents the results on the **Standard Stream**. Using a two-tailed paired t-test ($\alpha = 0.05$), we find that the experience replay baseline leads to significant reduction in Forgetting ($p = 0.016$) over lifelong learning. For the other metrics, however, experience replay does not lead to any significant improvements. Using the same t-test, we find that neither A-GEM nor MbPA++ perform significantly better than the experience replay baseline on any of the metrics. MbPA++, in particular, seems to have a high variance which might be due to the stochastic local adaptation procedure. The results indicate that, on our **Standard Stream**, these methods do not lead to a significant improvement over the experience replay baseline. In Appendix A.5, we further experiment with different experience replay rates and find that, as expected, the majority lead to a significant decrease in Forgetting and a significant increase in Final Accuracy compared to lifelong learning.

**In conclusion**, we find that the prevalent memory-based solutions to lifelong learning cannot outperform our simple experience replay baseline on any of the metrics. Although experience replay shows significant improvements in Forgetting and Final Accuracy, our general lifelong learning desiderata precludes its extensive use. The memory requirement of experience replay, parameterized by the write probability $P_{write}$, grows linearly with lifelong data stream while the space complexity desideratum calls for constraining the size of the memory buffer. Furthermore, the online stream desideratum necessitates limiting the replay interval and the number of replay examples. While sparse experience replay represents a simple and sound baseline, it is not an exemplar solution to lifelong learning. There is a need for more effective and realistic general lifelong learning solutions.

**Future work** could investigate prompt-based learning as a means to expanding degree-of-belief to generation tasks through auto-regressive blank-filling of cloze-style question prompts [16]. Recent advancements such as *pattern-exploiting training* (PET) [48] have been shown to perform well in few-shot learning of various NLP tasks [49]. Methods such as continuous task-specific vectors as defined in prefix-tuning [32] and AutoPrompt [51] can be used to build on the utilization of implicit task descriptors further.

**Broader impact** Advancements in lifelong learning will decrease unnecessary re-training of old models leading to reduced financial and environmental costs [1]. Our online stream desideratum encourages development of models suitable for privacy-sensitive scenarios where data (e.g., private medical records) are only used for training once without requiring to store them forever. We present our experimental framework as a unified and generalized benchmark suite that can facilitate comparative studies as well as the development of realistic models in this emerging field of research.

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
