# OpenReview forum: "Towards a robust experimental framework and benchmark for lifelong language learning"
_NeurIPS.cc/2021/Track/Datasets_and_Benchmarks/Round1 — NeurIPS 2021 Datasets and Benchmarks Track (Round 1)_

### Official Review · Reviewer_X3a5 · 2021-07-02
**Interesting Resource, but evaluation and paper need more work.**

**Rating:** 6
**Confidence:** 4

**Strengths:**

This paper takes a very interesting view of the problem of lifelong learning and, with the degree-of-belief setup, a very different approach from the existing benchmarks. It provides a very comprehensive set of arguments why a controlled environment is needed, and why existing environments fail to measure the performance of life-long learning and conflate multi-label and multi-class tasks. The paper clearly demonstrates that there is a benefit from using this environment over the existing ones.


**Weaknesses:**

Overall, the benchmark seems like a very promising resource, but its evaluation and the the paper itself have issues that I cannot overlook. I thus cannot give too high of a score. Most critically, many of the points I comment on below regard the seeming lack of any regard to results beyond the raw numbers.
Among others, there notably is lack of information how the included data points were selected, what the chosen selection strategy means for results, and how the test sets were constructed to provide informative analyses. Moreover, no broader impact was discussed at all.

Starting with the setup of the benchmark, I disagree with two of the outlined requirements:

1. I disagree with the requirement that there are no task identifiers, even if the same model is addressing all the tasks. If it turns out that the best performing model splits its parameters between tasks, then that is a useful empirical finding. Moreover, task identifiers have been very successfully used in models like T5.
2. Regarding requirement C, there is a difference between a model seeing a datapoint exactly once and a model being able to see it multiple times, especially when there is not enough training data to fully learn a task in the seeing-it-once regime. What about lifelong learning setups where a model is allowed to converge before moving on to a next task? This seems especially important since the entire benchmark is designed for low-resource (10k datapoints) regimes. How does this generalize to larger tasks?

My second point of critique regards the positioning of the contribution. The degree-of-belief setup fundamentally excludes all generation and retrieval tasks due to output sparsity. This restriction should be prominently featured in the paper as a drawback of the method instead of hidden in the paragraph starting at l232, especially since sequential learning is often investigated in multilingual generation settings like MT. Trying to cast this fundamental restriction as a benefit as the paper attempts in that paragraph seems extremely disingenuous when considering that it pitches as a testbed for lifelong learning for NLP and not “text problems that can be cast as dense classification problems”.

My third point concerns the nature of the benchmark as aiming to summarize a model in a single number. Equally important to measuring overall degradation of performance is to know which examples suffer the most, and I would expect a benchmark to provide this answer. However, AULTC has the same problem as simply measuring accuracy -- head-examples are overrepresented and thus, numbers are inflated. What should be separately measured is the effect on less representative examples in the data since those may be more likely to be forgotten. Even more dangerously, only a maximum of 1k test examples were selected for each task but no information on the process was provided. That means that it is likely to further the discrepancy outlined above.

The final flaw in the evaluation and presentation is that the discussion of results completely neglects the confidence intervals. The intervals suggest that many of the results are not significant, but are instead presents as fact.

**Additional Feedback:**

- As someone who mostly works on non-iid testing where the test distribution is different from the train distribution, I am wondering if you could discuss how the framework can handle a sequence of those tasks (so, doubly non-iid)
- What is the effect of task distances on the difficulty of learning in this setting. The paper describes orders of tasks but doesn't investigate similarities/differences further.

**Clarity:**

- Figure 4 is pixelated
- Q: How were the 10k training examples selected? What is the effect of this choice on the convergence of models? Do findings generalize?
- Q: How were the languages for the multilingual setup chosen?


**Correctness:**

As mentioned above, the confindence intervals were ignored in the interpretation of results.

**Documentation:**

- There is no link to the data sheets of the included datasets anywhere, only the documentation of the overall benchmark (which, as mentioned above, lacks crucial detail).
- For other points, see below.

**Ethics:**

- Table 4 fails to cite the creators of datasets not included in the final benchmark, despite the checklist answer to 4(a) that all creators were attributed.
- There is no broader / societal impact discussed at all in this paper when the whole reason of providing a ninth page is to more prominently feature this. Even a “we foresee no harmful impact from this work” would have been better than writing [N/A] in the checklist, and it suggests a lack of engagement with the question.


**Relation To Prior Work:**

- The D-of-B setting further has a lot of similarities to noise-contrastive estimation and probably should be extended in a similar way (pick correct out of n instead of binary label) to gain the benefit of NCE that it is equivalent to cross-entropy learning.


**Summary And Contributions:**

The paper introduces a lifelong learning benchmark comprising multiple NLU tasks, that enable the evaluation of techniques that combat catastrophic forgetting in a much more principled way. The evaluation is done through a degree-of-belief framework which casts all included tasks as a similar problem, which addresses the issue that often, multi-label and multi-class learning are conflated in the lifelong learning literature. In an initial evaluation, the paper uses the benchmark to evaluate popular approaches and find that they often can't beat simple baselines.

---

> ### Author Response · Authors · 2021-07-15
> **Updated paper with new evaluation experiments and detailed discussions**
>
> We thank the reviewer for their extensive review and suggestions. We address all the points below.
>
> “lack of any regard to results beyond the raw numbers”
>
> We have extended Section 5 with a more detailed discussion of the results along with several new experiments.
>
> “neglects the confidence intervals”
>
> We have calculated significance for all the comparative experiments and have now extended Section 5 with a more clear discussion (lines 414-420).
>
> “...how the test sets were constructed”
>
> We perform random sampling (lines 272, 311) but ensure that the class distribution matches that of the full train/test set. We also run further analyses to investigate performance on the full test set and find that it closely resembles the results on our test set (see Appendix A.6).
>
> “Regarding requirement C...How does this generalize to larger tasks?”
> “How were the 10k training examples selected? Do findings generalize?”
>
> We specifically chose tasks for which models can learn reasonably well with 10k training examples (lines 273). However, we recognise your point and follow your suggestion by running experiments using varying amounts of training data (line 275). We observe that our findings still generalize: the performance gap between lifelong learning and multi-task/single-task learning remains (lines 334--345; Table 5 in the appendix).
>
> “How were the languages for the multilingual setup chosen?”
> Please see line 296.
>
> “disagree with the requirement that there are no task identifiers”
>
> We are in agreement here. That is why we are distinguishing between explicit & implicit task identifiers. We will always require task identifiers in some form such that a model knows which task to perform on a given input. However, what is desirable is for the model to learn to understand implicit task descriptions and act accordingly, instead of using explicit task IDs that are manually hard-coded. Our new gradient overlap analyses suggest that models learn to identify tasks through the use of implicit task descriptors and utilize different parameter subsets to learn and perform the identified task (lines 380--399).
>
> “the positioning of the contribution”
>
> We agree this is a limitation and not a benefit, and have added a new paragraph to elaborate on this (lines 226--236). Our benchmark focuses on lifelong language understanding and have made this clearer. While one framework cannot cover the full range of diverse and distinct NLP tasks, we do hope that it will serve as a platform for future research, facilitating progress in lifelong learning for NLP.
>
> “effect on less representative examples”
>
> We now include performance analyses on the minority classes and less representative examples, and plot the F1 scores, true positive rates and true negative rates throughout lifelong learning (lines 366--371).
>
> “a maximum of 1k test examples were selected for each task”
>
> We run further analyses to investigate performance on the full test set and find that conclusions match those on our Standard stream test set (lines 312--314). We primarily use the Standard stream given runtime increases prohibitively with the increase in test set size (2 hours to run an experiment using our test set vs 51 on the full version).
>
> “how the framework can handle a sequence of those tasks (doubly non-iid)"
>
> This could be framed within a few-/zero-shot lifelong learning setting. This is a challenging and interesting direction that we leave for future work. However, we believe our framework can serve as a platform for further research in this area.
>
> “effect of task distances on the difficulty of learning”
>
> We have identified and utilized the most challenging sequences of tasks (Section 4.3) and therefore directly focus on the most challenging setting. In Section 5, we have added new observations concerning task similarity (lines 344, 347). Furthermore, our gradient overlap experiments (lines 380--399) are based on the intersection or overlap between old and the new sub-networks, where such overlap should depend on the similarity between old and new tasks. For streams with different tasks and similarities, we observe different subsets being used.
>
> “no link to the data sheets”
>
> We have thoroughly searched for the source documentation of the original datasets. However, most were created before the Datasheet paper was introduced, and so they do not appear to have any datasheets. As we are not the original creators of the datasets, we are not able to create such datasheets. We therefore refer to their original sources (papers, websites and/or readme files; Appendix B).
>
> “no broader impact discussed”
> Included a discussion of the broader impact (lines 433--438).
>
> “fails to cite the creators of datasets not included in the final benchmark”.
> We have added the citations for the datasets considered in Appendix B, Table 11. Thanks for pointing out the mistake.
>
> We would like to thank the reviewer again for their comments and suggestions which have visibly improved the paper.

---

> > ### Comment · Reviewer_X3a5 · 2021-07-20
> > **Thank you for the updates**
> >
> > Thank you for the substantive additional discussions that visibly improved this paper.
> >
> > I want to point out here as well that the lack of original data sheet does not prevent you from documenting your used datasets. I highly encourage you to write this documentation as it will help others work with your data. In case that answers are not included in the original paper, stating "information was not provided" is okay in these cases. However, all of us researchers have to work together to improve dataset documentation.

---

### Official Review · Reviewer_kuC3 · 2021-07-05
**A straightforward and reasonable adaptation of BERT-based models to unifying different tasks for lifelong language learning**

**Rating:** 6
**Confidence:** 2
**Correctness:** The benchmark design is complete and …
**Clarity:** The paper is quite clear and easy to …

**Strengths:**

This paper focuses on the problem of lifelong language learning, and presents an informative discussion of existing works in this area and points out their limitations. The true lifelong learning setting defined by the authors are reasonable and well motivates the construction of the proposed benchmarks.

The authors choose to use BERT to unify different tasks -- it is not technically novel, but it is a natural and clever choice, serving as a general and flexible framework for studying the lifelong learning problem.



**Weaknesses:**

The authors claim that their proposed Degree-of-Belief framework has the advantage of not leaking explicit task identifiers. However, under their framework, different tasks have strong indicators (such as certain word/symbol use and sentence structures) based on the construction of "statements" (figure 3). To what extent the model learns to identify different tasks and how this affects the final performances are not discussed.

In the experiments, the authors present a comprehensive set of benchmarks covering task orders, multi-domain, multilingual, etc., and make several observations of the results, but do not provide sufficient explanations.

**Additional Feedback:**

Please see the weaknesses section.

**Documentation:**

The proposed benchmarks are well documented.

**Relation To Prior Work:**

Pervious literature in lifelong learning is surveyed, while related works in pre-trained language models are not sufficient, such as T5 [1], where a single model is able to perform different NLP tasks by concatenating task-specific textual instructions to the input.

[1] Raffel, Colin, et al. "Exploring the Limits of Transfer Learning with a Unified Text-to-Text Transformer." Journal of Machine Learning Research 21 (2020): 1-67.

**Summary And Contributions:**

This paper proposes Degree-of-Belief framework for true general lifelong learning, where, as the authors define, task plurality, task generality, online stream and space complexity have certain restrictions, and the authors show that existing works in lifelong learning do not satisfy one or more of these conditions to be the true lifelong learning setting. The authors implement their framework using BERT models with various language tasks, and test different baseline models as well as two existing memory-based methods in different language learning settings such as multi-domain, multilingual and different linguistic hierarchy. The authors further propose a new metric AULTC for measuring accuracy throughout lifelong learning.

======post rebuttal=======

The authors' rebuttal addressed most of my concerns, and I am keeping my original ratings.

---

> ### Author Response · Authors · 2021-07-15
> **Added new experiments and rewritten Section 5 to incorporate reviewer's feedback**
>
> We thank the reviewer for their thorough review and valuable feedback.
>
> "The authors claim that their proposed Degree-of-Belief framework has the advantage of not leaking explicit task identifiers. However, under their framework, different tasks have strong indicators (such as certain word/symbol use and sentence structures) based on the construction of "statements" (figure 3). To what extent the model learns to identify different tasks and how this affects the final performances are not discussed."
>
> We note that in order to make sure that the implicit task identifiers cannot be hard-coded or memorized by the model, we use semantically similar but also syntactically different statements when encoding a task in our framework.
> However, to further ensure that explicit cues from implicit task identifiers are not inadvertently memorized by the model, we run an additional experiment (lines 371--379) where we increase the number of different statements when encoding a task in our framework (Appendix A.8). In the Standard stream, we remove the punctuation marks from some of the statement templates of the BoolQ, UDPOS and FewRel tasks; and replace keywords such as 'positive' with 'good' and 'negative' with 'bad' in some of the Yelp Review task statements. We find that this does not affect our conclusions, suggesting that the model does not solely rely on specific cues in the input statements when attempting to identify tasks through the use of these implicit statements.
>
> "In the experiments, the authors present a comprehensive set of benchmarks covering task orders, multi-domain, multilingual, etc., and make several observations of the results, but do not provide sufficient explanations."
>
> We have now extended the Experiments and Results section (Section 5) to include additional analyses as well as a more detailed discussion of the results (lines 341--345, 350--355, 417-418).

---

### Official Review · Reviewer_jx5k · 2021-07-05
**A rich framework and dataset for testing various aspects of language learning**

**Rating:** 7
**Confidence:** 2
**Clarity:** The paper is very well written.

**Strengths:**

In general the paper is very well written, the related work, experiments and analyses look sound, and the paper has a significant amount of worthy contributions. I assume that the explanations will be rather clear to experts in the field. The authors provide a lot of details regarding why they tackled this task, the method followed for developing their framework and metrics, and they provide a lot of data to analyze. The results discussed in the last section seem to demonstrate that the followed approach is valuable.

**Weaknesses:**

Not that this is necessarily a weakness, but the paper is not accessible to people that are not trained in machine learning. The experiments and discussion section is very dense, but due to the general density of the paper I think it is difficult to do much better.

**Additional Feedback:**

Some comments/questions:

- Line 224: Even if there are two (or any low number of) different options for the Statement, couldn't that knowledge be used to give away the task? The syntax and meanings of the statement will be quite different for each task: looking at figure 3, for instance only PoS tagging contains a colon; only Relation extraction contains three groups of words separated by dashes, only Boolean Question Answering ends with a question mark, only Sentiment Analysis will mention positive or negative.

- Section 4.3: all mentioned tasks (except question-answering) are related to language analysis; what would a statement look like for tasks with a generation component such as Machine Translation, simplification, paraphrasis, summarisation or simply Natural Language Generation (e.g. data-to-text, where the input is some structured representation and the output a text)? Would these very important NLP tasks fit your framework?

Style/formatting:
- Abstract: "open set" -> "open-set"?
- Section 3: there is a subsection 3.1 but no 3.2.
- Section 3.1: "absense" -> "absence".



**Correctness:**

The method looks sound, and I did not see obvious flaws in the experimental design.

**Documentation:**

The Appendix B (which, with the rest of the appendices, should have been separated from the main paper) contain all the information necessary for obtaining the dataset used in the experiments. A library is released on GitHub under an MIT license, which is open to future contributions, and the authors say that the test suite will be released on HuggingFace in the future.

**Ethics:**

No ethical issue is addressed in the paper, but since the datasets are derived from existing external sources, I expect the potential issues to be already handled on the source side.

**Relation To Prior Work:**

The authors clearly position themselves with respect to the existing work in Section 3, and they precisely present their contributions.

**Summary And Contributions:**

Score maintained after authors' answers.

=============

The paper present a framework for assessing how generic models can learn new tasks without forgetting about the previously learnt ones. As a non-expert in machine learning in general, I found the paper quite technical and difficult to follow, so the (modest) review below may be of limited usefulness, I apologize for this.

The authors present experiments using 16 different datasets that correspond to 10 different tasks; the tasks are reformulated as a Context (the input), a Statement about the Context, and a Truth value about that Statement. The tested models are expected to confirm or contradict the assigned truth value, and several aspects beyond the general accuracy are being evaluated for each model.  The paper overall looks quite complete and sound, and reports on a significant amount of work done; I think it is a very interesting and rich paper worth publishing.

---

> ### Author Response · Authors · 2021-07-15
> **Updated the paper to incorporate additional feedback**
>
> First of all, we would like to extend our thanks to the reviewer for their constructive review.
>
>
> “Even if there are two (or any low number of) different options for the Statement, couldn't that knowledge be used to give away the task? The syntax and meanings of the statement will be quite different for each task: looking at figure 3, for instance only PoS tagging contains a colon; only Relation extraction contains three groups of words separated by dashes, only Boolean Question Answering ends with a question mark, only Sentiment Analysis will mention positive or negative.”
>
> We note that in order to make sure that the implicit task identifiers cannot be hard-coded or memorized by the model, we use semantically similar but also syntactically different statements when encoding a task in our framework.
> However, to further ensure that explicit cues from implicit task identifiers are not inadvertently memorized by the model, we run an additional experiment (lines 371--378) where we increase the number of different statements when encoding a task in our framework (Appendix A.8). In the Standard stream, we remove the punctuation marks from some of the statement templates of the BoolQ, UDPOS and FewRel tasks; and replace keywords such as ‘positive' with ‘good' and ‘negative' with ‘bad' in some of the Yelp Review task statements. We find that this does not affect our conclusions, suggesting that the model does not solely rely on specific cues in the input statements when attempting to identify tasks through the use of these implicit statements.
>
>
> “all mentioned tasks (except question-answering) are related to language analysis; what would a statement look like for tasks with a generation component such as Machine Translation, simplification, paraphrasis, summarisation or simply Natural Language Generation (e.g. data-to-text, where the input is some structured representation and the output a text)? Would these very important NLP tasks fit your framework?”
>
> Our framework is designed for language understanding tasks. We agree that this is a limitation and have added a new paragraph in Section 4.1 to clearly elaborate on this. While one framework cannot cover the full range of diverse and distinct NLP tasks, we hope that our framework and benchmark will serve as a platform for future research and ways in which these can be extended and adapted to cover other types of tasks in NLP.
>
> We have updated the paper to incorporate all the additional feedback and corrected the formatting errors pointed out. We thank the reviewer again for the fruitful discussion.

---

### Decision · Program_Chairs · 2021-07-26

**Decision:**

Accept

**Comment:**

The proposed lifelong learning benchmark enables a significantly improved evaluation of techniques to combat catastrophic forgetting. However, there are two suggestions for the authors:

1) Regarding data sheets, the authors are encouraged to document the datasets to the best of their abilities even if the original authors did not release a datasheet. There may be a certain reluctance because you don't feel entitled to document other's data, but you can consider contacting the authors of the original datasets and ask them to (i) do the datasheet or (ii) validate a datasheet that you write.

2) Include a more thorough discussion of prompt-based learning.
In particular, another AC gave the following suggestions:
```
In lines 210, 211: "Multiple different tasks can be encoded into this framework by using textual task descriptions in the 211 statement x itself. Instead of using explicit task identifiers which need to be hard-coded manually"
I feel like this is pretty similar to prompt-based learning [1,2,3,4,5]  that is an active topic explored in NLP when using the large contextualized pre-trained models.
Considering that the authors also claim that: "Specifically, we choose to work with pre-trained, 60 contextualized language models considering they have been a mainstay of this field"
I think some literature on prompt-based learning should be covered in this work (For example, implicit task identifier could be regarded as a continuous prompt [4]).
Actually, if digging deeply along this direction, the limitation of the proposed framework (Line 226) could be potentially alleviated by combining prompt learning with seq2seq-based pre-trained models [6][7].
[1] It's Not Just Size That Matters: Small Language Models Are Also Few-Shot Learners
[2] How Context Affects Language Models' Factual Predictions
[3] AutoPrompt: Eliciting Knowledge from Language Models with Automatically Generated Prompts
[4] Prefix-Tuning: Optimizing Continuous Prompts for Generation
[5] Making Pre-trained Language Models Better Few-shot Learners
[6] UNIFIEDQA: Crossing Format Boundaries with a Single QA System
[7] All NLP Tasks Are Generation Tasks: A General Pretraining Framework
```